# Phytotherapy and Drugs: Can Their Interactions Increase Side Effects in Cancer Patients?

Sarah Allegra [1,*], Silvia De Francia [1], Francesca Turco [2], Ilenia Bertaggia [2], Francesco Chiara [1], Tiziana Armando [2], Silvana Storto [2] and Maria Valentina Mussa [2]

[1] Department of Clinical and Biological Sciences, Clinical Pharmacology Service Franco Ghezzo, University of Turin, San Luigi Gonzaga University Hospital, Regione Gonzole 10, 10043 Orbassano, Italy

[2] Department of Public Health, University of Turin, Città della Salute e della Scienza University Hospital, 10124 Turin, Italy

* Correspondence: sarah.allegra@unito.it; Tel.: +39-011-6705442

**Abstract:** Background: The use of herbs to treat illnesses was common in all historical eras. Our aim was to describe the phytotherapeutic substances that cancer patients use most commonly, and to determine whether their use can increase side effects. Methods: This was a retrospective and descriptive study conducted among older adults actively undergoing chemotherapy, admitted at the Oncology DH Unit (COES) of the Molinette Hospital AOU Città della Salute e della Scienza in Turin (Italy). Data collection was conducted through the distribution of self-compiled and closed-ended questionnaires during chemotherapy treatment. Results: A total of 281 patients were enrolled. Evaluating retching and sage consumption was statistically significant in multivariate analysis. The only risk factor for dysgeusia was chamomile consumption. Ginger, pomegranate, and vinegar use were retained as mucositis predictors. Conclusions: Phytotherapeutic use needs more attention in order to decrease the risks of side effects, toxicity, and ineffective treatment. The conscious administration of these substances should be promoted for safe use and to provide the reported benefits.

**Keywords:** phytotherapy; chemotherapy; toxicity

## 1. Introduction

Phytotherapy has been defined by the World Health Organization (WHO) as the medical discipline that allows the correct use, for preventive or curative purposes, of medicinal plants and their derivatives (phytotherapics or phytomedicaments), in relation to the pharmacological properties of their chemical constituents; we can consider phytomedicines as finished medicinal products, with labels, which contain active ingredients of single plants or the associations of plants. They also include juices, gums, lipid fractions, essential oils, and all other substances of this kind [1].

Phytotherapics, such as synthetic drugs, act on the human organism as well as the animal due to the chemicals they contain, which are responsible for pharmacological activities. These so-called phytomedicines are, for all intents and purposes, plant-based drugs, since they are aimed at carrying out a therapeutic action. Their effect depends on the nature and concentration of the pharmacologically active chemical constituents. Although for each vegetable source, some characteristic active ingredients have been roughly identified, to which a certain therapeutic action is ascribed; in the vegetable source, there are other extraordinary mines of complementary substances that contribute to modulating their action.

In certain instances, while having a therapeutic action, the chemicals found in plant material are not used in high enough concentrations to produce biological activity in vivo [2]. A plant material preparation that exerts the best pharmacological activity cannot be obtained and applied in clinical trials due to the lack of exact identification of chemical elements in plant preparations that are required to accomplish specific therapeutic action.

All clinical and interventional trials carried out for complex natural mixes are thought to be affected by these characteristics, which are regarded as a primary source of heterogeneity and differences in outcomes and a common source of ambiguity in their metanalyses [3]. The synergy between components may be a crucial component of efficacy [4,5]. Since no clear active principles have been identified, it is frequently hypothesized that a variety of bioactive elements may have contributed to the reported therapeutic outcomes, perhaps in a synergistic manner. However, there has not been any research performed to date that would support this theory for natural remedies. As a result, the synergism hypothesis is currently also thought to serve as a justification technique for natural products that lack active principles that have been discovered [6,7].

Over the last few years, an increase in phytotherapeutic substance use, such as the use of herbs, extracts, and homeopathic medicines to prevent or treat illnesses, has been reported [8]. Approximately 80% of the world's population uses traditional medicine for mild disturbances and illnesses [9]. In addition, in the last decade, the attention on healthy food highlights the beneficial role of a rich diet, particularly considering spicy and aromatic herbs such as turmeric, garlic, and ginger, which seem to have an anticancer, anti-inflammatory, and antioxidant effect [8]. However, phytotherapic, homeopathic, and alimentary substances are not risk-free, since they can interact with administered drugs, leading to adverse events or drug efficacy decrease, and, in addition, they can directly produce toxic effects on the organism [10–12]. Despite numerous scientific studies, there is a lack of awareness about the risks of interactions between natural products and synthetic drugs. Many people maintain the idea that natural substances can attenuate drug toxicity and that the more herbs are used, the more benefits there will be, particularly with regard to drug-related side effects [13]. The subjects most at risk are children and seniors who are self-medicating, as well as patients with comorbidities, oncological diseases, cardiovascular diseases, diabetes, chronic renal failure, human immunodeficiency virus, and drug or food allergies [14]. For this reason, it is important to remember that a number of variables might have an impact on the adverse effects that herbal medications may produce:

1.  Geriatric age: patients over the age of 60 have a lower tolerance to natural or synthetic medications, which doubles their risk of overdosing and adverse responses [15];
2.  Renal and hepatic insufficiency: it is not recommended to consume any phytotherapics that are processed by these organs, such as aristolochia and magnolia officinalis for the kidney and all plants containing pyrrolizidine alkaloids for the liver [16];
3.  Obesity: it is well recognized that having an excessive amount of body fat can change the way drugs operate in the body, either making them more or less effective [17];
4.  Diet: The majority of interactions happen at the point of absorption, particularly when food intake is sufficient to prevent medication contact with the intestinal mucosa or when there is calcium or iron present, since these substances bind to pharmaceuticals, reducing their adsorption [18–21].

Transcultural phytotherapy, which differs from naturally occurring compounds made synthetically, is gaining importance globally and has long been included in self-medication and medical practice [15,22]. Combined with conventional therapy, the use of phytodrugs is efficacious as adjunctive therapy to treat cancer pain, one of the most agonizing signs for people with advanced cancer. Eight out of ten patients with advanced cancer are thought to suffer moderate to severe pain, and about 55% of these patients experience persistent pain related to their disease [23]. Cancer-related pain may be caused by inflammation, organ invasion, or damage to the nerves. According to a recent analysis, 23% of adult cancer patients utilize herbal remedies: the use was significantly higher among female cancer patients than male cancer patients [24]. Another study supported the finding that female breast cancer patients prefer using herbal medicine over other types of treatment, followed by females with genital cancer, patients with tumors of the digestive system, and males with genital cancer [25,26].

To date, in the literature, there is a lack of evidence about the risks of pharmacodynamics and pharmacokinetic interactions between natural and synthetic drugs. These interactions could involve P glycoprotein (Pgp, MRP), an ATP-dependent xenobiotic efflux pump,

or cytochrome P450 (CYP) family enzymes, which metabolize different drugs [9,27–29]. The interactions could lead to drug inhibition or induction [9]. For example, cancer patients use drugs to attenuate symptoms and chemotherapy-related side effects; however, this could also lead to inadequate treatment [30].

Data reported in the literature indicate that St. John's Wort hyperforin isolated from leaf/flower mixtures is an inductor of CYP450 (particularly CYP3A4) and Pgp, causing a reduction in the efficacy of many tumor-treating drugs, such as irinotecan (up to 42% less), imatinib, paclitaxel, cyclophosphamide, and vincristine. In addition, the induction of Pgp caused a reduction in etoposide levels [11,27]. Some of the other substances used as foods or alternative medicines, including garlic, turmeric, and green tea, can interfere with many drugs at the level of the CYP450 family or Pgp [11,27,31,32]. In the treatment of tumors, clinical studies reported that garlic is an inductor of CYP3A4, and it caused reductions in the activities of etoposide, paclitaxel, vinblastine, and vincristine. Green tea inhibits the transport of irinotecan and its metabolites. Turmeric can cause the inhibition of CYP3A4, with a possible increase in the toxicity of drugs such as etoposide and paclitaxel. It also seemed to cause the inhibition of Pgp activity with an increase in plasma level of etoposide and, therefore, its toxicity. Phytotherapics not only influence chemotherapics and antimetabolite pharmacokinetics, but also other drugs. One example is valerian's interaction with hypericum, alprazolam, midazolam, and dextromethorphan, or ginseng with warfarin, midazolam, and caffeine [9].

With regard to food, different pharmacological interactions and alterations have been verified: for example, oranges, pomelos, and, particularly, grapes contain agents able to inhibit CYPs. Grape juice inhibits CYP3A4, affecting the pharmacokinetics of drugs such as midazolam, felodipine, and cyclosporine, with a possible increase in toxicity [33]. There is a known interaction between warfarin and vitamin K, which is present in broccoli and spinach and can cause alterations at the coagulation level, or between licorice and an antihypertensive drug [19]. In one study conducted in 2011, it was reported that tamoxifen, a drug used to treat breast cancer, interacts with sesame seeds [34].

One study conducted in Europe reported that 35.9% of cancer patients use complementary drugs: of these, 30%–55% were herbs [35]. The reason for this consumption was probably the need to attenuate the side effects of chemotherapy and symptoms correlated with tumors, but also to increase immune defenses and physical well-being, and for the reported anticancer properties of phytotherapics [35,36]. However, only 50% of patients using unconventional therapy reported this to their doctors [26,37,38].

The aim of this study was to evaluate the possible role of phytotherapics on gastrointestinal toxicity development in cancer patients undergoing different chemotherapy regimens.

## 2. Materials and Methods

This was a descriptive retrospective study carried out at the Oncology DH Unit (COES) of the Molinette Hospital AOU Città della Salute e della Scienza in Turin (Italy) between March 2016 and September 2017. Inclusion criteria were as follows: age ≥18 years old, ability to understand and to agree to this study, Italian language, and actively undergoing chemotherapy.

In order to achieve the set objectives, a database was built in which the data were codified and collected through the distribution of a self-completion questionnaire. The database was built by numerous experts, including nurses, pharmacologists, and oncologists. It was elaborated after considering certain data reported in the literature. The questionnaire was composed of nine multiple-choice questions designed for patients undergoing chemotherapy treatment. The first part consisted of patient demographic data, including age, gender, height, weight, cancer pathology, tumor, node and metastasis staging, chemotherapy treatment, and performance status (Karnofsky index or Eastern Cooperative Oncology Group scale [13]). The second part investigated drug consumption (corticosteroids, antiemetics, antipyretics, analgesics, antihypertensives, and statins), use of phytotherapy, consumption of certain foods, and the presence of gastrointestinal side effects over the last month.

Each patient provided written informed consent to fill out the questionnaire, and we received approval for this study from the Hospital Directors. Participation was voluntary, anonymous, and without compensation.

Data were analyzed using SPSS release 22 (SPSS Inc., Chicago, IL, USA). All the variables were tested for normality with the Shapiro–Wilk test. The correspondence of each parameter was evaluated with a normal or non-normal distribution, through the Kolmogorov–Smirnov test. Quantitative variables were described via median and interquartile range (IQR, quartile 1; quartile 3) if the variables were not normally distributed. Qualitative variables were described via frequencies and percentages. Any predictive power of the considered variables on side effects was finally evaluated through univariate and multivariate logistic (OR, odd ratio) regression analyses (IC, interval of confidence at 95%). Factors with *p*-value < 0.2 in univariate analysis were considered in multivariate analysis (level of statistical significance *p*-value < 0.05).

These data were routinely recorded during daily clinical practice as a quality assurance measure and to explore improvements in the quality of services; therefore, ethics committee approval was not required. Confidentiality was guaranteed in data collection, analysis, and dissemination phase, presenting the results in aggregate form.

## 3. Results

A total of 281 patients were enrolled; their demographical, clinical, and pharmacological characteristics are shown in Table 1. Sex-related frequency of phytotherapic use is shown in Figure 1.

**Table 1.** Demographic, clinical, and pharmacological description of the enrolled patients.

| Patients (N) | 281 |
|---|---|
| Age (median, (IQR)) | 66 (57–72) |
| Males | 170 (60.5) |
| Body Mass Index (median (IQR)) | 24.09 (21.69–27.38) |
| Cycles of chemotherapy (median, (IQR)) | 3.5 (2–5) |
| Chemotherapy regimen (N (%)) | |
| Folinic acid, 5-Fluorouracil, and Oxaliplatin | 33 (11.8) |
| Raltitrexed | 8 (2.8) |
| Oxaliplatin and Irinotecan | 4 (1.4) |
| Gemcitabine and Cisplatin | 56 (19.9) |
| Capecitabine and Oxaliplatin | 48 (17.1) |
| Raltitrexed and Oxaliplatin | 5 (1.8) |
| Irinotecan | 55 (19.6) |
| Paclitaxel | 6 (2.1) |
| Leucovorin, 5-Fluorouracil, and Irinotecan | 12 (4.3) |
| Gemcitabine | 1 (0.4) |
| Carboplatin, Pemetrexed, and Demcizumab | 3 (1.1) |
| Leucovorin, 5-Fluorouracil, Irinotecan, and Bevacizumab | 26 (9.3) |
| Cisplatin and Etoposide | 1 (0.4) |
| Paclitaxel and Carboplatin | 1 (0.4) |
| Carboplatin and Etoposide | 1 (0.4) |
| Methotrexate | 1 (0.4) |
| Nivolumab | 1 (0.4) |
| Cetuximab | 1 (0.4) |
| Carboplatin and Pemetrexed | 3 (1.1) |
| Cisplatin and Vinorelbine | 1 (0.4) |
| Fluorouracil, Leucovorin, and Bavacizumab | 1 (0.4) |
| Docetaxel | 2 (0.7) |
| Gemcitabine and Paclitaxel | 2 (0.7) |
| Ifosfamide | 2 (0.7) |
| Bleomycin, Etoposide, and Platinum | 3 (1.1) |

**Table 1.** *Cont.*

| Patients (N) | 281 |
|---|---|
| Cisplatin and Cetuximab | 2 (0.7) |
| Leucovorin, 5-Fluorouracil, Irinotecan, and Panitumumab | 1 (0.4) |
| Pemetrexed | 1 (0.4) |
| Coadministered drugs (N (%)) | |
| Cortisone | 124 (45.2) |
| Antiemetics | 146 (52) |
| Antipyretics | 88 (31.3) |
| Analgesics | 140 (49.8) |
| Antihypertensives | 107 (38.1) |
| Statins | 32 (11.4) |
| Phytotherapic use (N (%)) | 266 (94.7) |
| Male | 162 (95.3) |
| Female | 104 (93.7) |
| Phytotherapics (N (%)) | |
| Aloe | 47 (17.1) |
| Broccoli | 147 (52.3) |
| Chamomile | 82 (29.2) |
| Echinacea | 3 (1.1) |
| Eucalyptus | 9 (3.2) |
| Garlic | 179 (63.7) |
| Ginger | 87 (31) |
| Ginseng | 21 (7.5) |
| Goji Berry | 13 (4.6) |
| Grapefruit | 40 (14.2) |
| Hypericum | 2 (0.7) |
| Licorice | 46 (16.4) |
| Mint | 64 (22.8) |
| Oat | 21 (7.5) |
| Pineapple | 130 (46.3) |
| Pomegranate | 60 (21.4) |
| Propolis | 26 (9.3) |
| Sage | 115 (4.9) |
| Soy | 27 (9.6) |
| Tobacco | 25 (8.9) |
| Turmeric | 60 (21.4) |
| Valerian | 43 (15.3) |
| Vinegar | 169 (60.1) |
| Adverse Events (N (%)) | |
| Retching | 141 (50.2) |
| Vomit | 57 (20.3) |
| Mucositis | 81 (28.8) |
| Dysgeusia | 138 (49.1) |
| Stipsis | 108 (38.8) |
| Diarrhea | 98 (34.9) |

N, number; IQR, interquartile range (quartile 1; quartile 3); %, percentage.

The logistic regression analysis was carried out, including age, sex, BMI, number of chemotherapy cycles, primary pathology, presence of metastasis, concomitant drugs, use of phytotherapy, and consumption of certain foods. Evaluating retching, age ($p < 0.001$; odd ratio, OR = 0.944; confidence interval of 95%, IC95% [0.918; 0.971]), and sage consumption ($p = 0.025$; OR = 1.920; IC95% [1.087; 3.392]) resulted in being statistically significant in multivariate analysis. Instead, considering vomit and antihypertensive drug administration ($p = 0.045$; OR = 0.503; IC95% [0.257; 0.986]) was retained. The only risk factor for dysgeusia was chamomile ($p = 0.009$; OR = 2.039; IC95% [1.198; 3.472]) consumption. BMI ($p = 0.025$; OR = 1.920; IC95% [1.087; 3.392]), ginger ($p = 0.025$; OR = 1.920; IC95% [1.087; 3.392]), pomegranate ($p = 0.025$; OR = 1.920; IC95% [1.087; 3.392]), and vinegar ($p = 0.025$; OR = 1.920; IC95% [1.087; 3.392]) use were retained as mucositis predictors. Diarrhea results were

predicted only by age ($p = 0.005$; OR = 0.967; IC95% [0.945; 0.990]). Evaluating stipsis, no statistically significant results were obtained. The results obtained with the multivariate logistic regression analysis are shown in Table 2.

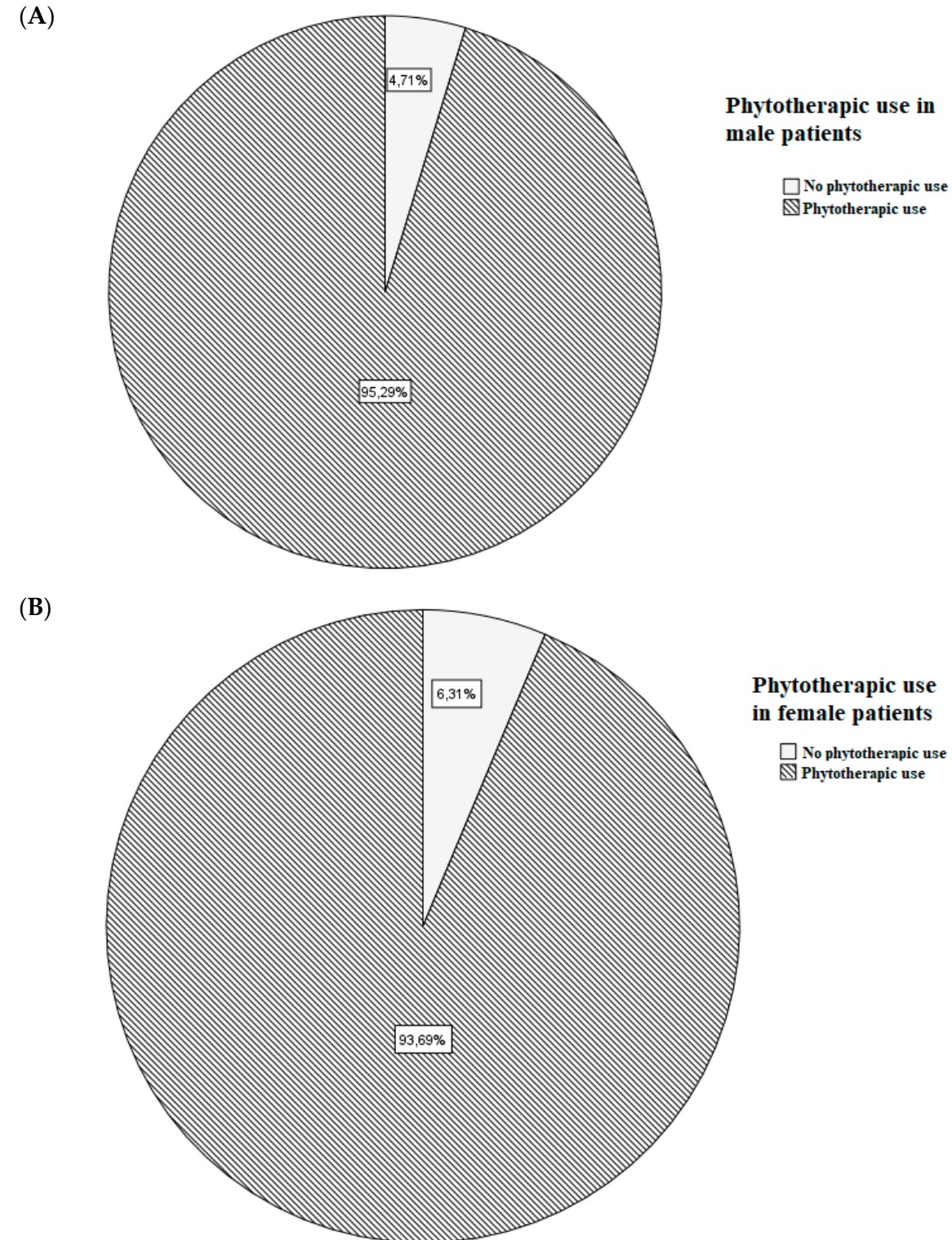

**Figure 1.** Frequency of phytotherapic use in male (**A**) and female (**B**) patients.

**Table 2.** Factors in multivariate logistic regression analyses able to predict retching, vomit, dysgeusia, mucositis, diarrhea, and stipsis.

| FACTOR | Retching | Vomit | Dysgeusia | Mucositis | Diarrhea | Stipsis |
|---|---|---|---|---|---|---|
| | *p* | *p* | *p* | *p* | *p* | *p* |
| | β (IC95%) | β (IC95%) | β (IC95%) | β (IC95%) | β (IC95%) | β (IC95%) |
| Age | <0.001 | 0.123 | 0.239 | | 0.005 | |
| | 0.944 (0.918–0.971) | 0.979 (0.954–1.006) | 0.987 (0.965–1.009) | | 0.967 (0.945–0.990) | |
| Gender | 0.413 | | | 0.066 | | |
| | 1.263 (0.722–2.210) | | | 1.691 (0.967) | | |
| BMI | | | | 0.018 | | |
| | | | | 1.074 (1.012–1.139) | | |
| Cortisone | | | 0.156 | | | |
| | | | 0.732 (0.476–1.126) | | | |
| Antiemetics | <0.001 | 0.001 | 0.024 | | | |
| | 3.317 (1.929–5.703) | 3.063 (1.599–5.870) | 1.736 (1.075–2.805) | | | |
| Antipyretics | 0.298 | 0.485 | | 0.594 | | |
| | 1.381 (0.752–2.536) | 1.274 (0.645–2.516) | | 1.184 (0.636–2.206) | | |
| Analgesics | 0.986 | 0.265 | | 0.180 | | |
| | 0.995 (0.554–1.787) | 1.423 (0.765–2.647) | | 1.463 (0.838–2.555) | | |
| Antihypertensives | | 0.045 | | | | |
| | | 0.503 (0.257–0.986) | | | | |
| Statins | 0.285 | | | | 0.289 | |
| | 0.791 (0.514–1.216) | | | | 0.755 (0.449–1.269) | |
| Ginger | 0.065 | 0.272 | | 0.048 | | |
| | 1.767 (0.966–3.235) | 1.439 (0.752–2.752) | | 1.784 (1.005–3.167) | | |
| Sage | 0.025 | | | | | |
| | 1.920 (1.087–3.392) | | | | | |
| Valerian | 0.101 | | | | | 0.061 |
| | 1.903 (0.881–4.112) | | | | | 1.895 (0.971–3.699) |
| Broccoli | | | | | 0.309 | |
| | | | | | 0.761 (0.451–1.287) | |

**Table 2.** *Cont.*

| | Retching | Vomit | Dysgeusia | Mucositis | Diarrhea | Stipsis |
|---|---|---|---|---|---|---|
| Tobacco | | | | | | 0.205 |
| | | | | | | 0.532 (0.201–1.410) |
| Echinacea | | | | 0.561 | | |
| | | | | 2.180 (0.158–30.109) | | |
| Ginseng | | | | 0.079 | | |
| | | | | 2.339 (0.906–6.035) | | |
| Propolis | 0.117 | | | 0.354 | | 0.094 |
| | 2.237 (0.817–6.121) | | | 1.554 (0.612–3.949) | | 2.049 (0.885–4.744) |
| Garlic | | | | | | 0.102 |
| | | | | | | 1.559 (0.915–2.655) |
| Pomegranate | | | | 0.003 | | |
| | | | | 2.658 (1.386–5.100) | | |
| Licorice | 0.566 | | | | 0.084 | |
| | 1.252 (0.581–2.694I) | | | | 1.822 (0.922–3.602) | |
| Aloe | | | | 0.193 | | |
| | | | | 1.605 (0.787–3.274) | | |
| Pineapple | 0.426 | | | | | |
| | 1.264 (0.710–2.250) | | | | | |
| Chamomile | | 0.149 | 0.009 | | | 0.073 |
| | | 1.597 (0.845–3.017) | 2.039 (1.198–3.472) | | | 1.643 (0.955–2.826) |
| Vinegar | | 0.335 | | | 0.010 | |
| | | 0.731 (0.387–1.382) | | | 0.465 (0.260–0.831) | |
| Goji Berry | | | | | 0.105 | |
| | | | | | 0.276 (0.058–1.310) | |

$p$: $p$ value; β: β coefficient; IC95%: interval of confidence at 95%.

## 4. Discussion

One of the oldest types of treatment on the planet is herbal medicine. Independent systems of plant-based treatment have evolved throughout history: Ayurveda in India, Kampo medicine in Japan, Sa-sang in Korea, and traditional Chinese medicine [39]. In 1804, the discovery of morphine acted as the trigger for the rational discovery of drugs from plants [40,41]. Later, pure compounds replaced crude extracts and partially refined natural products [42]. Natural products, on the other hand, have become less significant as a result of the field of chemistry's rapid advancement in the last few centuries, which encouraged high-throughput screens of synthetic compound libraries for drug development [40]. The

analysis of Newman and Cragg suggests that one-third of the drugs approved by the Food and Drug Administration between 1981 and 2014 were based on natural products, which is supported by the fact that fewer drugs have been approved as a result of the limited chemical diversity of synthetic compounds [43,44].

In cancer patients, the use of defined phytotherapeutic products to relieve symptoms or for the prevention of other diseases seems to be strongly recommended or strongly discouraged based on the type of chemo-adjuvant therapy assumed, especially from the prescription provided by the oncologist [45–50]. There is currently greater attention needed regarding monitoring the safety of these phytotherapeutic products. The WHO appointed a commission of experts to set the objectives and criteria on the basis of this, from the initial plant product to the final one [1,9]. However, it was already recognized that some plant compounds could induce adverse reactions.

Sex and age have previously been found to be crucial variables for the use of phytotherapy [51]. One explanation may be that women, in general, seem to be more aware of their health status: several studies have found that women are more likely than men to take care of their health status, and they use more natural supplements than men [44–47]. While the use of phytopharmaceuticals decreases with age, the use of prescribed medications increases: the results showed that the likelihood of taking herbal medicines, alone or in combination with other medications or food supplements, declined with age [46,47]. In our cohort, we observed that males (162 of 170 patients) used phytotherapics more frequently than females (104 of 111 patients); however, this data were not in accordance with the literature [43,52,53]. Women and men respond differently to each kind of treatment: this mainly depends on physiological, anatomical, and hormonal characteristics. The existence of differences between drug and food pharmacokinetics and pharmacodynamics influences the response to treatments. Although this was already known since 1932, the year in which the first report on the gender difference in the pharmacology of barbiturates in rats was reported, full awareness of the relevance of the role of gender pharmacology only came at the end of the last century. By pharmacokinetics, we mean the study of the four phases of a drug's transition in our body: absorption, distribution, metabolism, and elimination. These four stages are primarily influenced by age and hormones, thus showing significant differences related to sex. Pharmacodynamics, on the other hand, indicates the effect of a therapeutic agent on bodies and studies the biochemical and physiological effects and their mechanism of action. There are numerous pharmacodynamic differences depending on sex, mainly mediated by hormones, genes, and the environment; however, the pharmacokinetic differences are simpler to analyze, and the pharmacodynamic differences are more difficult to detect [54,55]. However, both should deserve a worthy study in terms of gender differences, or the resulting effect will be limited and approximate.

In general, in our study, 94.7% of patients used at least one phytotherapic, including food. Sage, chamomile, ginger pomegranate, and vinegar resulted in gastrointestinal side effect risk factors (Table 3).

**Table 3.** List of phytotherapics resulting in risk factors of gastrointestinal side effects.

| Phytotherapic | | Side Effect |
|---|---|---|
| Sage | *Salvia officinalis* L. | retching |
| Chamomilla | *Matricaria chamomilla* L. | dysgeusia |
| Ginger | *Zingiber officinale* | mucositis |
| Pomegranates | *Punica granatum* L. | mucositis |

In the second stage of the formalin test, an extract of sage (*Salvia officinalis L.*) demonstrated a dose-dependent anti-inflammatory effect. In a mouse model of neuropathy, the sage hydroethanolic extract was also able to reduce pain induced by vincristine [56]. These receptors may be involved in the antinociceptive actions of sage since diverse species of the *Salvia genus* contain a diterpene termed salvinorin A, an agonist of opioid receptors [57]. By preventing DNA damage and caspase-3 overactivation, rosmarinic acid from sage extracts

is similarly helpful in preventing the death of brain cells [58]. A recent review on breast cancer and integrative medicine used to reduce disease symptoms reported that liquid extracts or tinctures of essential oils of sage can reduce mucositis and stomatitis [59]. With our analysis, we observed a higher risk of retching in those who took sage.

Traditional medical practices in many nations have employed *Matricaria chamomilla L* to treat a variety of illnesses, including respiratory issues, liver disorders, neuropsychiatric issues, common colds, and gastrointestinal disorders. Additionally, this plant is frequently used to treat disorders of the skin, eyes, and mouth, as well as to treat pain and infections [60]. Through the use of several experiments, the antioxidant activity of chamomile essential oil and extracts was examined: catalase, acetylcholine esterase, glutathione, peroxidase, ascorbate peroxidase, and superoxide dismutase enzyme activity of extracts were also evaluated. Additionally, studies were carried out on the antioxidant activity found in cell suspension culture, waste extracts from chamomile processing, and polyphenolic–polysaccharide conjugates [61–63]. The antibacterial ability of M. chamomilla, on the other hand, was demonstrated against both Gram-positive and Gram-negative bacteria [63]. Moreover, chamomile is involved in many interactions; for example, it inhibits many CYP isoforms, such as CYP3A4, CYP2D6, and CYP2C9, with a possible increase in etoposide, paclitaxel, and cyclophosphamide toxicity. It seems to interact with drugs' antioestrogenic activity [64]. In addition, this plant has an antioxidant, antimicrobial and anti-inflammatory action and it has been used as an oral mucositis treatment [65,66]. Here, we observed chamomile's predictive role on dysgeusia symptoms.

Ginger (*Zingiber officinale*) is used as traditional medicine worldwide. Different models have been used to conduct scientific investigations on the ethnomedical applications of African ginger's usefulness against asthma, sinusitis, colds, flu, malaria, and other inflammation-related illnesses. In addition, a variety of bacterial and fungal species were tested against the antibacterial activity of African ginger extracts [67]; the anti-inflammatory ginger potential effect is based on cyclooxygenase (COX-1 and -2) inhibitory activity [68]. Prostaglandin-synthesis inhibitors, glucocorticoids, and histamine receptors acting as mediators of inflammation are among potential molecular targets for antiasthmatic or anti-allergic medications. Ginger ethanoic extracts were discovered to be a powerful inhibitor of COX enzymes' ability to synthesize prostaglandins [69–71]. Ginger-based modulation of nuclear factor kappa-(NF-K), which controls the transcription and expression of various pro-inflammatory cytokines, was assumed to be the mechanism of pain and other inflammation-related symptom alleviation. The diethyl ether extract from oven-dried rhizome also demonstrated good activity in the glucocorticoid receptor binding and phosphodiesterase IV enzyme assays, which is an indication of its potential anti-inflammatory and antiasthmatic properties, respectively, according to Fouche et al. [72]. Numerous studies have been carried out on ginger, a substance known for a long time for its antiemetic activities and, therefore, often used to counteract the side effects of chemotherapy. In fact, it seems that ginger contains a wide range of bioactive compounds (in particular gingerol and shogaol) that can act on multiple mechanisms involved in the onset of nausea and vomiting induced by chemotherapy. Properties include the ability to act on the 5-HT3 receptor, substance P, to antagonize the acetylcholine receptor, and to affect gastrointestinal motility and gastric emptying rate [20]. Furthermore, it contains 6-gingerol and 6-shogaol, which are able to relieve 5-fluorouracil-induced oral mucositis and pain. The mechanism is probably related to the regulation of $Na^+$ channels [73]. On the contrary, we reported a higher risk of mucositis related to ginger use. Probably, this difference could be related to concomitant chemotherapy administration. A thorough analysis of randomized controlled trials revealed that there is still insufficient evidence to endorse ginger's clinical application. Moreover, research on lung cancer patients, who were receiving cisplatin-based chemotherapy regimens and were given ginger for the treatment of chemotherapy-induced nausea, revealed that ginger did not have any advantages over a placebo. Participants reported a higher incidence of both acute and delayed nausea; however, when given concurrently with aprepitant (an antiemetic medicine), it sometimes worsened the degree

of nausea. Due to its ability to speed up gastric emptying and increase intestinal motility, ginger may lessen medication effectiveness and gastrointestinal absorption. Contrary to widespread assumption, it is not suggested to utilize it as a chemical to be added to standard chemotherapy treatments [20,74].

Important polyphenolic chemicals found in pomegranates (*Punica granatum L.*), such as ellagitannins and punicalagin, have potent antioxidant properties that can scavenge free radicals and produce metal chelates in biological tissues. According to in vitro and in vivo research, pomegranates' anti-inflammatory and antioxidant qualities have significant antimutagenic and antiproliferative effects that control gene expression, modify cellular processes, and reduce the ability of cancer to spread. Pomegranate components may be useful for the prevention and treatment of cancer, particularly colorectal and prostate cancer, according to a small number of clinical studies [75]. Pomegranate peel is composed of punicalagin, which is extracted into pomegranate juice in large quantities and represents the bioactive constituent responsible for pomegranates' antioxidant activity [76]. Moreover, it is able to elicit apoptosis in colorectal cancer cells by cytochrome c release and caspase 9 and caspase 3 stimulation [77]. Chen and colleagues reported the potential colorectal cancer and chemotherapy-induced intestinal mucositis therapy of pomegranates [78]. We confirm these results by reporting pomegranates' positive predictive role on mucositis.

Due to its properties, vinegar has medical uses in addition to being used as a food condiment. In traditional Chinese medicine, it is used as a purgative because it potentially soothes the liver, reduces depression, avoids blood stasis, and relieves pain [79]. Due to its multiple health advantages, including its antihyperglycemic, antihypercholesterolemic, antihypertension, antimicrobial, antithrombotic, and anticancer actions, vinegar has recently received a lot of attention. In recent years, interest in vinegar's favorable metabolic effects has grown as a result of its positive effects. Acetic acid, organic acids, amino acids, peptides, vitamins, and mineral salts are all present in vinegar [80,81]. Vinegar's healthy beneficial role is mediated by the oxidation of low-density lipoproteins inhibition, leading to antidiabetic effects and lowering blood cholesterol levels [82,83]. Acetic acid diffuses through microorganisms' cell membranes, leading to bacterial cell death. Moreover, the bioactive compounds of vinegar, such as polyphenols and vitamins, may reduce the incidence of degenerative illnesses [84]. In addition, we observed a higher risk of mucositis linked to vinegar consumption.

## 5. Conclusions

In conclusion, can the use of herbal medicines increase the toxic effects in patients treated with chemotherapy? It depends. From the results of our study, it seems that some plant products are able to predict or increase the risk of certain adverse effects. However, this retrospective study has several limitations: first, the concomitant use of two or more types of vegetal products has not been considered; second, the dose of each product has not been evaluated in this analysis. Thus, further studies, including more detailed information about the use of phytotherapics in terms of concomitant use and dose, are needed to confirm the obtained results.

To date, the idea that natural substances have only healthy effects is still very diffused [9,13]. Particularly, subjects at risk are self-medicating patients with comorbidities, oncological and cardiovascular diseases, diabetes, chronic renal failure, human immunodeficiency virus infections, and drug or food allergies [14]. However, phytotherapics and foods interact with organisms and synthetic drugs, but this does not always lead to benefits. For this reason, the literature reported natural compound-related toxic effects [8,11,12,31,85]. A lot of variables can affect this mechanism: for example, chemotherapy type, concomitant drugs, and diet. Additionally, the involvement of the gut microbiota in the biotransformation of xenobiotics consumed by humans has received a lot of attention: it seems to have a substantial impact on the potency and safety of both natural and synthetic medications [86]. The conscious administration of these substances should be promoted, for safe use and in order to provide the reported benefits. The field of phytotherapy is still unknown to many

health professionals; therefore, greater dissemination of information is necessary, so that the patient can be adequately assisted in any kind of assistance; moreover, further studies are needed to broaden the knowledge on complications and assistance, in such a way as to bring about a development of this area towards better quality standards.

**Author Contributions:** Conceptualization, S.D.F. and M.V.M.; methodology, T.A. and S.S.; software, S.A.; validation, S.D.F., T.A. and S.S.; formal analysis, F.T. and I.B.; data curation, F.T. and I.B.; writing— original draft preparation, S.A.; writing—review and editing, F.C.; supervision, S.D.F. and M.V.M.; project administration, M.V.M. All authors have read and agreed to the published version of the manuscript.

**Funding:** This research received no external funding.

**Institutional Review Board Statement:** Ethical review and approval were waived for this study due to these data being routinely recorded during daily clinical practice as a quality assurance measure and to explore improvements in the quality of services; therefore, ethics committee approval was not required. Confidentiality was guaranteed in the data collection, analysis, and dissemination phase, presenting the results in aggregate form.

**Informed Consent Statement:** Informed consent was obtained from all subjects involved in this study.

**Data Availability Statement:** Not applicable.

**Conflicts of Interest:** The authors declare no conflict of interest.

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
