# Peer review of "Phytotherapy and Drugs: Can Their Interactions Increase Side Effects in Cancer Patients?"

_jox, doi:10.3390/jox13010007_

Round 1
Reviewer 1 Report
In this research, Authors evaluates the possible relationship between food supplement consumption and and an increase of side effects in cancer patients undergoing chemotherapy. The topic is interesting because it highlights a still underestimated problem related to the use of self prescribed phytotherapeutics which, although of natural origin, may contain several active ingredients whose interactions with other drugs (not only chemotherapics) are not always clear.
Statistical methods and sample size are approriate
Minor revisions:
English should be revised
Author Response
In this research, Authors evaluates the possible relationship between food supplement consumption and and an increase of side effects in cancer patients undergoing chemotherapy. The topic is interesting because it highlights a still underestimated problem related to the use of self prescribed phytotherapeutics which, although of natural origin, may contain several active ingredients whose interactions with other drugs (not only chemotherapics) are not always clear.
Statistical methods and sample size are appropriate
R: Thank you for your comment.
Minor revisions:
English should be revised
R: Thank you for your suggestion, a full text revision has been made.
Reviewer 2 Report
Dear Authors,
review is interesting and quite novel, however it feels like the initial question at title is not clearly answered, especially at conclusion section (Can Their Interactions Increase Side Effects in Cancer Patients?)
A self-completion questionnaire was used, it would be nice to present even as supplement.
I suggest to add a table with all plants used and Traditional medical practicesmentioned at discussion section, so it will be easy for readers to check.
Minor spelling and grammar mistakes, most references are too old, more than 10 years old, need update.
major revision is needed.
Author Response
Dear Authors,
review is interesting and quite novel, however it feels like the initial question at title is not clearly answered, especially at conclusion section (Can Their Interactions Increase Side Effects in Cancer Patients?)
R: Thank you for your comment, the conclusion section has been modified.
A self-completion questionnaire was used, it would be nice to present even as supplement.
R: Thank you for the suggestion, the questionnaire has been added as supplementary material.
I suggest to add a table with all plants used and Traditional medical practicesmentioned at discussion section, so it will be easy for readers to check.
R: The table (Table 3) has been added in conclusion section, has you required.
Minor spelling and grammar mistakes, most references are too old, more than 10 years old, need update.
R: Thank you for your suggestion, a full text revision has been made.
Reviewer 3 Report
jox-2040327, Phytotherapy and Drugs: Can Their Interactions Increase Side Effects in Cancer Patients?
The manuscript is interesting and has a good value. Even if the study is small it could have an impact for future similar studies. The authors should try better to have an objective view on their data. A major problem is that they consider that the natural products used don’t interact between them. The study consider that the patients use only one type of vegetal product, and not a complex mix. Also, they authors seem to ignore the doses of vegetal products used. Their analysis is a yes or no type, but surely the patients used different doses of the same product. The readers of the article should be made aware of these problems and I strongly advise the authors to present these limitations in the final paper.
Row 53, the authors should also add that the phytotherapic, homeopathic and alimentary substances can directly produce toxic effects on the organism.
Row 68, please add the plant tissue you are referring to.
In table 1, the authors present Folinic acid and also Leucovorin. As far as I know, it is the same thing. Also, use one single way to present a drug, Fluorouracil or 5-Fluorouracil. Not both!
Mesna is not really a chemotherapic drug
The order of plants in table 1 can be improved. List them alphabetically. For mucositis, check the numbers!
Figure 1 and 2 can be joined in a single one with 2 panels.
Row 159, explain what IC95 and OR means
Row 247, it should be Zingiber officinale, and not Siphonochilus aethiopicus
Section 299 to 307 is out of place. I think it is left from another manuscript. The authors should be more careful and really read all the paper.
Author Response
The manuscript is interesting and has a good value. Even if the study is small it could have an impact for future similar studies. The authors should try better to have an objective view on their data. A major problem is that they consider that the natural products used don’t interact between them. The study consider that the patients use only one type of vegetal product, and not a complex mix. Also, they authors seem to ignore the doses of vegetal products used. Their analysis is a yes or no type, but surely the patients used different doses of the same product. The readers of the article should be made aware of these problems and I strongly advise the authors to present these limitations in the final paper.
R: Thank you for your professional comment. The limitations section has been added.
Row 53, the authors should also add that the phytotherapic, homeopathic and alimentary substances can directly produce toxic effects on the organism.
R: Thank you for your suggestion, the sentence has been changed.
Row 68, please add the plant tissue you are referring to.
R: The required information has been inserted, thank you.
In table 1, the authors present Folinic acid and also Leucovorin. As far as I know, it is the same thing. Also, use one single way to present a drug, Fluorouracil or 5-Fluorouracil. Not both!
R: I apologize for the mistake, the table has been corrected.
Mesna is not really a chemotherapic drug
R: Correct, the indication has been removed. Thank you!
The order of plants in table 1 can be improved. List them alphabetically. For mucositis, check the numbers!
R: Thank you for the suggestions. The name of plants has been listed alphabetically and the number of mucositis has been corrected.
Figure 1 and 2 can be joined in a single one with 2 panels.
R: The figures have been marged in a single figure with 2 panels.
Row 159, explain what IC95 and OR means
R: The definitions of OR and IC95% have been inserted, thank you for the revision.
Row 247, it should be Zingiber officinale, and not Siphonochilus aethiopicus
R: Thank you for the professional suggestion, the sentence has been changed.
Section 299 to 307 is out of place. I think it is left from another manuscript. The authors should be more careful and really read all the paper.
R: The section has been removed.
Round 2
Reviewer 2 Report
Authors replied to all my comments adequately and adapted all my suggestions. Their manuscript has been now improved and it can be published in its current form.